# On the Statistical Properties of Multiscale Permutation Entropy: Characterization of the Estimator’s Variance

**DOI:** 10.3390/e21050450

**Published:** 2019-04-30

**Authors:** Antonio Dávalos, Meryem Jabloun, Philippe Ravier, Olivier Buttelli

**Affiliations:** Laboratoire Pluridisciplinaire de Recherche en Ingénierie des Systèmes, Mécanique, Énergétique (PRISME), University of Orléans, 45100 Orléans, INSA-CVL, France

**Keywords:** Multiscale Permutation Entropy, ordinal patterns, estimator variance, Cramér–Rao Lower Bound, finite-length signals

## Abstract

Permutation Entropy (PE) and Multiscale Permutation Entropy (MPE) have been extensively used in the analysis of time series searching for regularities. Although PE has been explored and characterized, there is still a lack of theoretical background regarding MPE. Therefore, we expand the available MPE theory by developing an explicit expression for the estimator’s variance as a function of time scale and ordinal pattern distribution. We derived the MPE Cramér–Rao Lower Bound (CRLB) to test the efficiency of our theoretical result. We also tested our formulation against MPE variance measurements from simulated surrogate signals. We found the MPE variance symmetric around the point of equally probable patterns, showing clear maxima and minima. This implies that the MPE variance is directly linked to the MPE measurement itself, and there is a region where the variance is maximum. This effect arises directly from the pattern distribution, and it is unrelated to the time scale or the signal length. The MPE variance also increases linearly with time scale, except when the MPE measurement is close to its maximum, where the variance presents quadratic growth. The expression approaches the CRLB asymptotically, with fast convergence. The theoretical variance is close to the results from simulations, and appears consistently below the actual measurements. By knowing the MPE variance, it is possible to have a clear precision criterion for statistical comparison in real-life applications.

## 1. Introduction

Information entropy, originally defined by Shannon [1], has been used as a measure of information content in the field of communications. Several other applications of entropy measurements have been proposed, such as the analysis of physiological electrical signals [2], where a reduction in entropy has been linked to aging [3] and various motor diseases [4]. Another application is the characterization of electrical load behavior, which can be used to perform non-intrusive load disaggregation and to design smart grid applications [5].

Multiple types of entropy measures [6,7,8] have been proposed in recent years to assess the information content of time series. One notable approach is the Permutation Entropy (PE) [9], used to measure the recurrence of ordinal patterns within a discrete signal. PE is fast to compute and robust even in the presence of outliers [9]. To better measure the information content at different time scales, Multiscale Permutation Entropy (MPE) [10] was formulated as an extension of PE, by using the multiscale approach proposed in [11]. Multiscaling is particularly useful in measuring the information content in long range trends. The main disadvantage of PE is the necessity of a large data set for it to be reliable [12]. This is especially important in MPE, where the signal length is reduced geometrically at each time scale. Signal-length limitations have been addressed and improved with Composite MPE and Refined Composite MPE [13].

PE theory and properties have been extensively explored [14,15,16]. However, there is still a lack of understanding regarding the statistical properties of MPE. In our previous work [17], we already derived the expected value of the MPE measurement, taking into consideration the time scale and the finite-length constraints. We found the MPE expected value to be biased. Nonetheless, this bias depends only on the MPE parameters, and it is constant with respect to the pattern probability distribution. In practice, the MPE bias does not depend on the signal [17], and thus, can be easily corrected.

In the present article, our goal is to continue this statistical analysis by obtaining the variance of the MPE estimator by means of Taylor series expansion. We develop an explicit equation for the MPE variance. We also obtain the Cramér–Rao Lower Bound (CRLB) of MPE, and compare it to our obtained expression to assess its theoretical efficiency. Lastly, we compare these results with simulated data with known parameters. This gives a better understanding of the MPE statistic, which is helpful in the interpretation of this measurement in real-life applications. By knowing the precision of the MPE, it is possible to take informed decisions regarding experimental design, sampling, and hypothesis testing.

The reminder of the article is organized as follows: In Section 2, we lay the necessary background of PE and MPE, the main derivation of the MPE variance, and the CRLB. We also develop the statistical model to generate the surrogate data simulations for later testing. In Section 3, we show and discuss the results obtained, including the properties of the MPE variance, its theoretical efficiency, and similarities with our simulations. Finally, in Section 4, we add some general remarks regarding the results obtained.

## 2. Materials and Methods

In this section, we establish the concepts and tools necessary for the derivation of the MPE model. In Section 2.1 we review the formulation of PE and MPE in detail. In Section 2.2, we show the derivation of the MPE variance. In Section 2.3, we derive the expression for the CRLB of the MPE, and we compare it to the variance. Finally, in Section 2.4, we explain the statistical model used to generate surrogate signals, which are used to test the MPE model.

### 2.1. Theoretical Background

#### 2.1.1. Permutation Entropy

PE [9] measures the information content by counting the ordinal patters present within a signal. An ordinal pattern is defined as the comparison between the values of adjacent data points in a segment of size *d*, known as the embedded dimension. For example, for a discrete signal of length *N*, $x_{1},\ldots ,x_{n},\ldots ,x_{N}$, and d=2, only two possible patterns can be found within the series, $x_{n}<x_{n+1}$ and $x_{n}>x_{n+1}$. For d=3, there are six possible patterns present, as shown in Figure 1. In general, for any integer value of *d*, there exists *d* factorial (d!) possible patterns within any given signal segment. To calculate PE, we must first obtain the sample probabilities within the signal, by counting the number of times a certain pattern i=1,…,d! occurs. This is formally expressed as follows:(1)$$P\left(\pi_{i}\right)=\frac{♯\{n|n\leq N-\left(d-1\right),\left(x_{n},\ldots ,x_{n+d-1}\right)\;type\;\pi_{i}\}}{N-(d-1)}={\hat{p}}_{i}.$$where $\pi_{i}$ is the label of a particular ordinal pattern *i*, and ${\hat{p}}_{i}$ is the estimated pattern probability. Some authors [15] also introduce a downsampling parameter in Equation (Equation 1), to address the analysis of an oversampled signal. For the purposes of this article, we assume a properly sampled signal, and thus, we do not include this parameter.

Using these estimations, we can apply the entropy definition [1] to obtain the PE of the signal
(2)$$\hat{H}=\frac{-1}{\ln(d!)}\sum_{i=1}^{d!}{\hat{p}}_{i}\ln{\hat{p}}_{i}.$$
where $\hat{H}$ is the estimated normalized PE from the data.

PE is a very simple and fast estimator to calculate, and it is invariant to non-linear monotonous transformations [15]. It is also convenient to note that we need no prior assumptions on the probability distribution of the patterns, which makes PE a very robust estimator. The major disadvantage in PE involves the length of the signal, where we need N≫d! for PE to be meaningful. This imposes a practical constraint in the use of PE for short signals, or in conditions where the data length is reduced.

#### 2.1.2. Multiscale Permutation Entropy

The MPE consists of applying the classical PE analysis on coarse-grained signals [10]. First, we define *m*, as the time scale parameter for the MPE analysis. The coarse-graining procedure consists in dividing the original signal into adjacent, non-overlapping segments of size *m*, where *m* is a positive integer less than *N*. The data points within each segment are averaged to produce a single data point, which represents the segment at the given time scale [11].
(3)$$x_{k}^{\left(m\right)}=\frac{1}{m}\sum_{j=m(k-1)+1}^{km}x_{j}.$$

MPE consists in applying the PE procedure in Equation (Equation 2) on the coarse-grained signals in Equation (Equation 3) for different time scales *m*. This technique allows us to measure the information inside longer trends and time scales, which is not possible using PE. Nonetheless, the resulting coarse-grained signals have a length of N/m, which limits the analysis. Moreover, for a sufficiently large *m*, the condition N/m≫d! is eventually not satisfied.

At this point, it is important to discuss some constraints regarding the interaction between time scale *m* and the signal length *N*. First, strictly speaking, N/m must be an integer. In practice, the coarse-grained signal length will be the integer number immediately below N/m. Second, the proper domain of *m* is (0,N), as the segments size, at most, can be the same length as the signal itself. It is handy to use a normalized scale $\frac{m}{N}$ with domain (0,1) which does not change between signals. This normalized scale is the inverse of the coarse-grained signal length. Taking the length constraints from the previous section, this means that $\frac{m}{N}\ll \frac{1}{d!}$. For d=2, a normalized scale of $\frac{m}{N}=\frac{1}{2}$ will result in a coarse-grained signal of 2 data points, which is not meaningful for this analysis. For d=3, the normalized scale must be significantly less than $\frac{1}{6}$, which corresponds to a coarse-grained signal of six elements. Therefore, for practical applications, we restrict our analysis for values of $\frac{m}{N}$ close to zero, by selecting a small time scale *m*, a large signal length *N*, or both.

### 2.2. Variance of MPE Statistic

The calculated MPE can be interpreted as a sample statistic that estimates the true entropy value, with an associated expected value, variance, and bias, for each time scale. We have previously developed the calculation of the expected value of the MPE in [17], where the bias has been found to be independent from the pattern probabilities. We expand on this result by first proposing an explicit equation to the MPE statistic, and then we formulate the variance of MPE, as a function of the true pattern probabilities and time scale.

For the following development, we use *H*, the non-normalized version of Equation (Equation 2), for convenience.
(4)$$\hat{H}=\hat{H}\ln\left(d!\right)=-\sum_{i=1}^{d!}{\hat{p}}_{i}\ln{\hat{p}}_{i}.$$

By taking the Taylor expansion of Equation (Equation 4) on a coarse-grained signal in Equation (Equation 3), we get
(5)$$\hat{H}=H\;-\;\frac{m}{N}\sum_{i=1}^{d!}\left(1+ln\left(p_{i}\right)\right)\Delta Y_{i}\;-\;\sum_{k=1}^{\infty }\frac{{\left(-1\right)}^{k+1}}{k(k+1)}{\left(\frac{m}{N}\right)}^{k+1}\sum_{i=1}^{d!}\frac{{\Delta Y_{i}}^{k+1}}{p_{i}^{k}}$$
where $\hat{H}$ is the MPE estimator, *H* is the true unknown MPE value, *m* is the time scale, *N* is the signal length, and *d* the embedded dimension. The probabilities $p_{i}$ correspond to the true probabilities (unknown parameters) of each pattern, and $\Delta Y_{i}$ correspond to the random part in the multinomially (Mu) distributed frequency of each pattern,
(6)$$\begin{array}{c} Y_{i}=\frac{N}{m}{\hat{p}}_{i}=\frac{N}{m}p_{i}+\Delta Y_{i},\\ \left\{Y_{1},\ldots ,Y_{d!}\right\}∼Mu\left(\frac{N}{m},p_{1},\ldots ,p_{d!}\right) \end{array}$$
$Y_{i}$ being the number of pattern counts of type *i* in the signal.

If we take into consideration the length constraints discussed in Section 2.1.2, we know that the normalized time scale $\frac{m}{N}$ is very close to zero. This implies that the high-order terms of Equation (Equation 5) will quickly vanish for increasing values of *k*. Therefore, we propose to make the simplest approximation of Equation (Equation 5) by taking only the term k=1. By doing this, we get the following expression: (7)$$\hat{H}\;\approx \;H\;-\;\frac{m}{N}\sum_{i=1}^{d!}\left(1+ln\left(p_{i}\right)\right)\Delta Y_{i}\;-\;\frac{1}{2}{\left(\frac{m}{N}\right)}^{2}\sum_{i=1}^{d!}\frac{{\Delta Y_{i}}^{2}}{p_{i}}.$$

Using our previous results involving MPE [17], we know the expected value of Equation (Equation 7) is
(8)$$E\left[\hat{H}\right]\approx H-\frac{d!-1}{2}\left(\frac{m}{N}\right).$$

The statistic presents a bias that does not depend on the pattern probabilities $p_{i}$, and thus, can be easily corrected for any signal.

Now, we move further and calculate the variance of the MPE estimator. First, it is convenient to express Equation (Equation 4) in vectorial form.
(9)$$\hat{H}=-\hat{l}{\;}^{T}\hat{p}$$
where
(10)$$\begin{array}{cc} \hat{p} & ={\left[{\hat{p}}_{1},\ldots ,{\hat{p}}_{d!}\right]}^{T}\\ \hat{l} & ={\left[\ln\left({\hat{p}}_{1}\right),\ldots ,\ln\left({\hat{p}}_{d!}\right)\right]}^{T}. \end{array}$$
$\hat{p}$ being the column vector of pattern probability estimators, $\hat{l}$ the column vector of the logarithm of each pattern probability estimator, and *T* is the transpose symbol. We can now rewrite Equation (Equation 7) as
(11)$$\hat{H}\;\approx \;H\;-\;\frac{m}{N}{\left(1+l\right)}^{T}\Delta Y\;-\;\frac{1}{2}{\left(\frac{m}{N}\right)}^{2}{\left(p^{\circ -1}\right)}^{T}\Delta Y^{\circ 2}$$
where
(12)$$\begin{array}{cc} p & ={\left[p_{1},\ldots ,p_{d!}\right]}^{T}\\ 1 & ={\left[1,\ldots ,1\right]}^{T}\\ p^{\circ 2} & ={\left[p_{1}^{2},\ldots ,p_{d!}^{2}\right]}^{T} \end{array}\;\begin{array}{cc} p^{\circ -1} & ={\left[p_{1}^{-1},\ldots ,p_{d!}^{-1}\right]}^{T}\\ l & ={\left[\ln\left(p_{1}\right),\ldots ,\ln\left(p_{d!}\right)\right]}^{T}\\ \Delta Y^{\circ 2} & ={\left[\Delta Y_{1}^{2},\ldots ,\Delta Yp_{d!}^{2}\right]}^{T}. \end{array}$$

The circle notation ∘ represents the Hadamard power (element-wise).

Now, we can obtain the variance of the MPE estimator (Equation 11),
(13)$$\begin{array}{cc} var(\hat{H}) & =E\left[{\hat{H}}^{2}\right]-{\left(E[\hat{H}]\right)}^{2}\\ \approx & H^{2}\;+\;{\left(\frac{m}{N}\right)}^{2}{\left(1+l\right)}^{T}E\left[\Delta Y\Delta Y^{T}\right]\left(1+l\right)\;-\;{\left(\frac{m}{N}\right)}^{2}{\left(p^{\circ -1}\right)}^{T}E\left[\Delta Y^{\circ 2}\right]H\\ & +{\left(\frac{m}{N}\right)}^{3}{\left(1+l\right)}^{T}E\left[\Delta Y{\left(\Delta Y^{\circ 2}\right)}^{T}\right]\left(p^{\circ -1}\right)\;+\;\frac{1}{4}{\left(\frac{m}{N}\right)}^{4}{\left(p^{\circ -1}\right)}^{T}E\left[\Delta Y^{\circ 2}{\left(\Delta Y^{\circ 2}\right)}^{T}\right]\left(p^{\circ -1}\right)\\ & +\left(\frac{m}{N}\right)\left(d!-1\right)H\;-\;\frac{1}{4}{\left(\frac{m}{N}\right)}^{2}{\left(d!-1\right)}^{2}\;-\;H^{2}. \end{array}$$

Now, we know that $E\left[\Delta Y\Delta Y^{T}\right]$ is the Covariance matrix of ΔY, which is the multinomial random variable defined in Equation (6). The matrix $E\left[\Delta Y{\left(\Delta Y^{\circ 2}\right)}^{T}\right]$ is the Coskewness matrix, and $E\left[\Delta Y^{\circ 2}{\left(\Delta Y^{\circ 2}\right)}^{T}\right]$ is the Cokurtosis. We know that (see Appendix A),
(14)$$\begin{array}{cc} E\left[\Delta Y\Delta Y^{T}\right] & =\frac{N}{m}\left(diag\left(p\right)-pp^{T}\right) \end{array}$$
(15)$$\begin{array}{cc} E\left[\Delta Y{\left(\Delta Y^{\circ 2}\right)}^{T}\right] & =2\frac{N}{m}\left(p^{\circ 2}p^{T}-diag\left(p^{\circ 2}\right)\right)+\frac{N}{m}\left(diag\left(p\right)-pp^{T}\right) \end{array}$$
(16)$$\begin{array}{cc} E\left[\Delta Y^{\circ 2}{\left(\Delta Y^{\circ 2}\right)}^{T}\right] & =3\frac{N}{m}\left(\frac{N}{m}-2\right)p^{\circ 2}{\left(p^{\circ 2}\right)}^{T}\;-\;\frac{N}{m}\left(\frac{N}{m}-2\right)\left(p^{\circ 2}p^{T}+p{\left(p^{\circ 2}\right)}^{T}\right)\\ & +{\left(\frac{N}{m}\right)}^{2}pp^{T}\;-\;4\frac{N}{m}\left(\frac{N}{m}-2\right)diag\left(p^{\circ 3}\right)\;+\;2\frac{N}{m}\left(\frac{N}{m}-3\right)diag\left(p^{\circ 2}\right)\\ & +\frac{N}{m}\left(diag\left(p\right)-pp^{T}\right) \end{array}$$
where diag(p) is a diagonal matrix, where the diagonal elements are the elements of p.

We can further summarize the covariance matrix (14) as follows:(17)$$\frac{N}{m}\left(diag\left(p\right)-pp^{T}\right)=\frac{N}{m}\Sigma_{p}.$$

We also rewrite the following term in Equation (Equation 13),
(18)$${\left(p^{\circ -1}\right)}^{T}E\left[\Delta Y^{\circ 2}\right]=\left(\frac{N}{m}\right){\left(p^{\circ -1}\right)}^{T}\left(p-p^{\circ 2}\right)=\left(\frac{N}{m}\right)\left(d!-1\right).$$

By combining Equations (14) to (Equation 18) explicitly in Equation (Equation 13), we get the expression,
(19)$$var\left(\hat{H}\right)\;\approx \;\left(\frac{m}{N}\right)l^{T}\Sigma_{p}l\;+\;{\left(\frac{m}{N}\right)}^{2}\left(1^{T}l+d!H+\frac{1}{2}\left(d!-1\right)\right)\;+\;\frac{1}{4}{\left(\frac{m}{N}\right)}^{3}\left(1^{T}p^{\circ -1}-\left(d{!}^{2}+2d!-2\right)\right).$$

We note that Equation (Equation 19) is a cubic polynomial respect to the normalized scale m/N. Since we are still working in the region where m/N is close to (but not including) zero, we propose to further simplify this expression using only the linear term. This means that we can approximate Equation (Equation 19) as follows:(20)$$\begin{array}{cc} var(\hat{H}) & \approx \;\left(\frac{m}{N}\right)l^{T}\Sigma_{p}l\;=\;\frac{m}{N}\left(\sum_{i=1}^{d!}p_{i}\ln^{2}\left(p_{i}\right)-\sum_{i=1}^{d!}\sum_{j=1}^{d!}p_{i}p_{j}\ln\left(p_{i}\right)\ln\left(p_{j}\right)\right)\;=\;\frac{m}{N}\left(\sum_{i=1}^{d!}p_{i}\ln^{2}\left(p_{i}\right)-H^{2}\right). \end{array}$$

We note that Equation (Equation 20) will be equal to zero for a perfectly uniform pattern distribution (which yields a maximum PE). In this particular case, Equation (Equation 20) will not be a good approximation for the MPE variance, and we will need, at least, the quadratic term in Equation (Equation 19). Except for extremely high or low values of MPE, we expect the variance linear approximation to be accurate. We discuss more properties of this statistic in the Section 3.2.

### 2.3. MPE Cramér–Rao Lower Bound

In this section, we compare the MPE variance (Equation 20) to the CRLB, to test the efficiency of our estimator. The CRLB is defined as
(21)$$CRLB\left(H\right)=\frac{{\left[1-B^{\prime }\left(H\right)\right]}^{2}}{I(H)}\leq var\left(\hat{H}\right)$$
where *B* is the bias of the MPE expected value from Equation (Equation 8) and
(22)$$\begin{array}{cc} B^{\prime }\left(H\right) & =\frac{dB}{dH}=-\frac{d}{dH}\left(\frac{d!-1}{2}\frac{m}{N}\right)=0\\ I(H) & =-E\left[\frac{\partial^{2}\ln\left(f_{H}\left(y;p\right)\right)}{\partial H^{2}}\right] \end{array}$$
where I(H) is the Fisher Information, and $f_{H}\left(y;p\right)$ is the probability distribution function of *H*.

Although we do not have an explicit expression for $f_{H}$, we can express CRLB(H) as a function of CRLB(p) as follows [18]:(23)$$CRLB\left(H\right)={\left(\frac{\partial H}{\partial p}\right)}^{T}CRLB\left(p\right)\frac{\partial H}{\partial p}$$
where
(24)$$\begin{array}{cc} \frac{\partial H}{\partial p} & ={\left[\frac{\partial H}{\partial p_{1}},\ldots \frac{\partial H}{\partial p_{d!}}\right]}^{T} \end{array}$$
(25)$$\begin{array}{cc} CRLB(p) & =I{\left(p\right)}^{-1}\leq cov_{p}\left(\hat{p}\right). \end{array}$$
I(p) is the Fisher information matrix for p, which we know has a multinomial distribution related to Equation (6). Each element of I(p) is defined as
(26)$$I_{j,k}\left(p\right)=-E\left[\frac{\partial^{2}}{\partial p_{j}\partial p_{k}}\ln\left(f_{\hat{p}}\left(\hat{p};p\right)\right)\right].$$
where $f_{\hat{p}}\left(\hat{p};p\right)$ is the probability distribution of $\hat{p}$, which is identically distributed to Equation (6) (for the full calculation of CRLB(H), see Appendix B). Thus, by obtaining the inverse of I(p) and all partial derivatives of *H* with respect to each element of p, we obtain the CRLB(H).
(27)$$CRLB\left(H\right)=\frac{m}{N}\left(\sum_{i=1}^{d!}p_{i}\ln^{2}p_{i}-H^{2}\right)=\frac{m}{N}l^{T}\Sigma_{p}l.$$

As we recall from our results in Equations (Equation 19) and (Equation 20), the CRLB corresponds to the first term of the Taylor series expansion of the MPE variance. The high-order terms in Equation (Equation 19) increase the MPE variance above the CRLB. For small values of $\frac{m}{N}$, the higher-order terms in Equations (Equation 20) become neglectable, which make the MPE variance approximation converge to CRLB(H).

### 2.4. Simulations

To test the MPE variance, we need an appropriate benchmark. We need to design a proper signal model with the following goals in mind: First, the model must preserve the pattern probabilities across all the signal generated; second, the function must have the pattern probability as an explicit parameter, easily modifiable for testing. The following equation satisfies these criteria for dimension d=2:(28)$$x_{n}=x_{n-1}+\epsilon_{n}-\delta \left(p\right)$$
where
(29)$$\begin{array}{cc} p & =\frac{1}{2}\left(1-erf\left(\frac{\delta (p)}{\sqrt{2}}\right)\right) \end{array}$$
(30)$$\begin{array}{cc} \delta (p) & =\sqrt{2}\;erf^{-1}\left(1-2p\right). \end{array}$$

This is a non-stationary process, with a trend function δ(p). ϵ is a Gaussian noise term with variance $\sigma^{2}=1$, without loss of generality. Although different values of $\sigma^{2}$ will indeed modify the trend function, it will not affect the pattern probabilities in the simulation, as PE is invariant to non-linear monotonous transformations [15].

For dimension d=2, $p=p_{1}=P\left(x_{n}<x_{n+1}\right)$ and $1-p=p_{2}=P\left(x_{n}>x_{n+1}\right)$, which are the probabilities of each of the two possible patterns. The formulation of δ(p) comes from the Gaussian Complementary Cumulative Distribution function, taking *p* as a parameter. This guarantees that we can directly modify the pattern probabilities *p* and 1−p for simulation.

Although *p* is not invariant at different time scales, it is consistent within each coarse-grained signal, which suffices for our purposes. We chose to restrict our simulation analysis to the embedded dimension d=2. Although our theoretical work holds for any value of *d* (see Section 2.2), it is difficult to visualize the results at higher dimensions.

This surrogate model (Equation 28) was implemented in the Matlab environment. For the test, we generated 1000 signals each, for 99 different values for p=0.01,0.02,…0.99. The signal length was set to N=1000. Some sample paths are shown in Figure 2. These signals were then subject to the coarse-graining procedure (Equation 3) for time scale m=1,…,10. The MPE measurement was taken for each coarse-grained simulated signal using Equation (Equation 2). Finally, we obtained the mean and variance of the resulting MPE’s for each scale. This simulation results are discussed in Section 3.2.

## 3. Results and Discussion

In this section, we explore the results from the theoretical MPE variance. In Section 3.1, we contrast the results from the model against the MPE variance measured from the simulations. In Section 3.2, we discuss these findings in detail.

### 3.1. Results

Here, we compare the theoretical results with the surrogate data obtained by means of the procedure described in Section 2.4. We use the cubic model from Equation (Equation 19) instead of the linear approximation in Equation (Equation 20), to take in consideration non-linear effects that could arise from simulations. Figure 3 shows the theoretical predictions in dotted lines, and simulation measurements in solid lines.

Figure 3a shows the variance of the MPE (Equation 19) as a function of *p* for d=2. The lines correspond to a normalized time scale of m/N=0.001,0.005, and 0.010. The variance presents symmetry with respect to the middle value of p=0.5. This is to be expected, as for d=2, the variance has only one degree of freedom. The structure is preserved, albeit scaled, for different values of *m*.

Figure 3b–d show the MPE variance versus the normalized time scale, for different values of pattern probability *p*. As we can see in Figure 3b, the variance increases linearly with respect to the normalized scale m/N at the positive vicinity of zero, as predicted in Equation (Equation 20). This is not the case for when p=0.5 (maximum entropy), as shown in Figure 3c, where both the theory and simulations show a clear non-linear tendency. Finally, Figure 3d shows the case where we have extreme pattern probability distributions, with entropy close to zero. Although we use the complete cubic model (Equation 19), the predicted curves are almost linear.

In general, we can observe that the simulation results have greater variance than the prediction of the model. The real values from the simulations correspond to the sample variance from the signals, calculated from $\hat{p}$ instead of p. Nonetheless, the simulations and theoretical graphs have the same shape. It is interesting to note that the discrepancies increase with the scale. This effect is addressed in Section 3.2.

### 3.2. Discussion

It is interesting to explore the particular structure of the variance. As we can observe from Figure 3b, the MPE variance increases linearly with respect to the time scale for a wide range of pattern distributions, as described in Equation (Equation 20). This is even true with highly uneven distributions, where the entropy is very close to zero, as shown in Figure 3d. Nonetheless, when we observe the expression (Equation 20), the equation collapses to zero when all pattern probabilities $p_{i}$ are the same. For any embedded dimension *d*, all probabilities $p_{i}=\frac{1}{d!}$. This particular pattern probability distribution is the discrete uniform distribution, which yield to the maximum possible entropy in Equation (Equation 2). As we can observe from Figure 3c, the linear approximation in Equation (Equation 19) is not enough to estimate the variance in this case. Nonetheless, the results suggest a quadratic increase. This agrees with previous results by Little and Kane [16], where they characterized the classical normalized PE variance for white noise under finite-length constraints.
(31)$$var\left(\hat{H}\right)\approx \frac{d!-1}{2{\left(\ln d!\right)}^{2}}\frac{1}{N^{2}}$$

This result matches our model in Equation (Equation 19) (taking the quadratic term) for the specific case of uniform pattern distribution and time scale m=1. This suggest that, when we approach the maximum entropy, the quadratic approximation is necessary.

Contrary to the bias in the expected value of MPE [17], the variance is strongly dependent on the pattern probabilities present in the signal, as shown in Figure 3a. For the MPE with embedded dimension d=2, the variance of MPE has a symmetric shape around equally probable patterns. We observe, for a fixed time scale, that the variance increases the further we deviate from the center, and sharply decreases for extreme probabilities. The variance presents its lowest points at the center and the extremes of the pattern probability distributions, which corresponds to the points where the entropy is maximum and minimum, respectively. For d=2, we can calculate the variance (Equation 20) more explicitly,
(32)$$var\left(\hat{H}\right)\approx \left(\frac{m}{N}\right)l^{T}\Sigma_{p}{l|}_{d=2}\;=\;\left(\frac{m}{N}\right)p\left(1-p\right)\ln^{2}\left(\frac{p}{1-p}\right).$$

It is evident that the zeros of this equation correspond to p=0, p=1 (points of minimum entropy), and p=0.5 (maximum entropy). We can get the critical points by calculating the first derivative of Equation (Equation 32) with respect to *p*
(33)$$\left(\frac{m}{N}\right)\frac{d}{dp}\left(l^{T}\Sigma_{p}{l|}_{d=2}\right)=\left(\frac{m}{N}\right)\ln\left(\frac{p}{1-p}\right)\left(\left(1-2p\right)\ln\left(\frac{p}{1-p}\right)+2\right)=0$$
which is equal to zero to get the extreme points. From the first term of Equation (Equation 33), we obtain the critical point p=0.5, which is a minimum. If the second term of the equation is equal to zero, we need to solve the transcendental function
(34)$$\ln\left(\frac{p}{1-p}\right)=\frac{2}{2p-1.}$$

Numerically, we found the maximum points to be p=0.083 and p=0.917, as shown in Figure 3a. Both these values yield to a normalized PE value of $\hat{H}=0.413$. This implies that, when we obtain values of MPE close to this value, we will have a region with maximum variance. This effect arises directly from the pattern probability distribution. Hence, in the region around $\hat{H}=0.413$ for d=2, we will have maximum estimation uncertainty, even before factoring the finite length of the signal, or the time scale. Therefore, Equation (Equation 20) implies an uneven variance across all possible values of the entropy measurement, regardless of time scale and embedded dimension. It also implies a region where this variance is maximum.

Lastly, as noted in Section 2.3, the MPE variance (Equation 20) approaches the CRLB for small values of *m*. This is further supported by the simulation variance MPE measurements shown in Figure 3, which are consistently above the theoretical prediction. We can attribute this effect to the number of iterations of the testing model in Equation (Equation 28), where an increasing number of repetitions will yield to a more precise MPE estimation, with a reduced variance. Since we already include the effect of the time scale in Equations (Equation 19) and (Equation 20), the difference between the theoretical results and the simulations does not come directly from the signal length or the pattern distribution.

## 4. Conclusions

By following on from our previous work [17], we further explored and characterized the MPE statistic. By using a Taylor series expansion, we were able to obtain an explicit expression of the MPE variance. We also derived the Cramér–Rao Lower bound of the MPE, and compared it to our obtained expression. Finally, we proposed a suitable signal model for testing our results against simulations.

By analyzing the properties of the MPE variance graph, we found the estimator to be symmetric around the point of equally probable patterns. Moreover, the estimator is minimum in both the points of maximum and minimum entropy. This implies, first, that the variance of the MPE is dependent on the MPE estimation itself. In regions where the entropy measurement is near the maximum or minimum, the estimation will have a minimum variance. On the other hand, there is an MPE measurement where the variance will have a maximum. This effect comes solely from the pattern distribution, and not from the signal length or the MPE time scale. We should take in account this effect when comparing real entropy measurements, as it could affect the interpretation of statistically significant difference.

Regarding the time scale, the MPE variance linear approximation is sufficiently accurate for almost all pattern distributions, provided that the time scale is small compared to the signal length. An important exception to this is the case where the pattern probability distribution is almost uniform. For equally probable patterns, the linear term of the MPE variance vanishes, regardless of time scale. In this case, we need to increase the order of the approximation to the quadratic term. Hence, for MPE values close to the maximum, the variance presents a quadratic growth respect to scale.

We found the MPE variance estimator obtained in this article to be almost efficient. When the time scale is small compared to the signal length, the MPE variance resembles the MPE CRLB closely. Since the CRLB is equal to the first term on the Taylor series approximation for the variance, this implies that the efficiency limit also changes with the MPE measurement itself. Since this effect also comes purely from the pattern distribution, we cannot correct it with the established improvements of the MPE algorithm, like Composite MPE or Refined Composite MPE [13].

By knowing the variance of the MPE, we can improve the interpretation of this estimator in real-life applications. This is important because researchers can impose a precision criterion over the MPE measurements, given the characteristics of the data to analyze. For example, the electrical load behavior analysis can be achieved using short-term measurements where the time scale is a limiting factor for MPE. By knowing the variance, we can compute the maximum time scale until which the MPE is still significant.

By better understanding the advantages and disadvantages of the MPE technique, it is possible to have a clear benchmark for statistically significant comparisons between signals at any time scale.

## Figures and Tables

**Figure 1 entropy-21-00450-f001:**
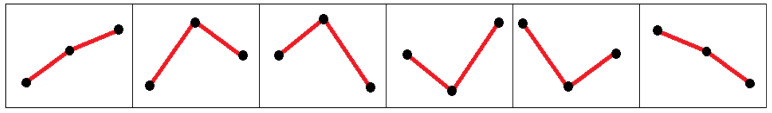
All possible patterns for embedded dimension d=3, from a three-point sequence $\left\{x_{n},x_{n+1},x_{n+2}\right\}$. The patterns represented, from left to right, are $\pi_{1}:\;x_{n}<x_{n+1}<x_{n+2}$, $\pi_{2}:\;x_{n}<x_{n+2}<x_{n+1}$, $\pi_{3}:\;x_{n+2}<x_{n}<x_{n+1}$, $\pi_{4}:\;x_{n+1}<x_{n}<x_{n+2}$, $\pi_{5}:\;x_{n+1}<x_{n+2}<x_{n}$, and $\pi_{6}:\;x_{n+2}<x_{n+1}<x_{n}$. The difference in amplitude between data points does not affect the pattern, as long as the order is preserved.

**Figure 2 entropy-21-00450-f002:**
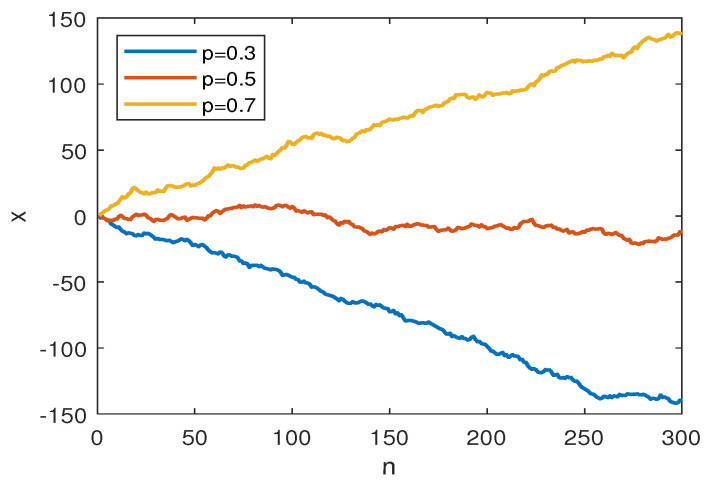
Simulated paths for MPE testing, where *p* is the probability of $x_{n}<x_{n+1}$ for dimension d=2. The graph shows sample paths for p=0.3, p=0.5, p=0.7

**Figure 3 entropy-21-00450-f003:**
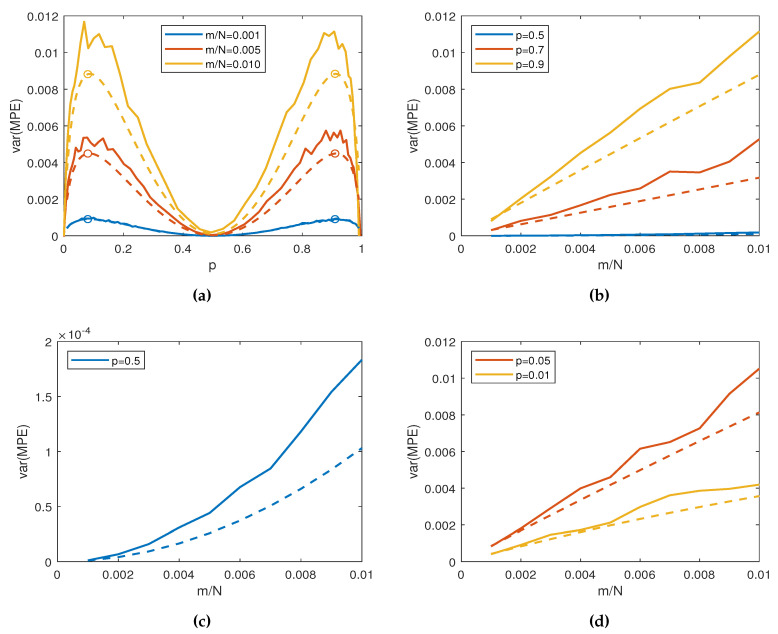
$var(\hat{H})$ for embedded dimension d=2 from theory (dotted lines) and simulations (solid lines). (**a**) $var(\hat{H})$ vs. pattern probability *p* for different normalized scales m/N. (**b**) $var(\hat{H})$ vs. m/N for different values of *p*. (**c**) $var(\hat{H})$ vs. m/N at p=0.5, which corresponds to maximum entropy. (**d**) $var(\hat{H})$ vs. m/N at with small *p*, which approaches minimum entropy.

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
