# Peer review of "On the Statistical Properties of Multiscale Permutation Entropy: Characterization of the Estimator’s Variance"

_entropy, 2019, doi:10.3390/e21050450_

Round 1

Reviewer 1 Report

The paper describes interesting analytical and experimental results to characterise the variance of (multiscale) permutation entropy (MPE and PE, respectively). I would expect that this results would be useful in the design of experiments and the anlaysis of (M)PE results, and they add value on top of recent papers that have been characterising the behaviour of the distributions of permutation patterns and of (M)PE for diverse types of signals.

In the following, I include some comments about the manuscript.

In Permutation Entropy (PE), sometimes an oversampling parameter (for example, "d") is included so that one sample is included in the permutation patterns out of each "d" consecutive ones. This may be useful in cases when the signal under analysis is severely oversampled. I appreciate that the manuscript focuses on a simplified description assuming d=1 but it would be useful to mention this choice explicitly.

Line 78: There is probably a typo in tfracNm.

After line 79, it would be useful to comment on the potential implications of limiting the series to k=1.

I would recommend reorganising the "Results and Discussions" in that Section 3.1 should in reality be part of the Methods as it covers the description of the simulations.

Author Response

Thank you very much for your comments and suggestions. We have edited the manuscript to better help the flow of ideas. We have also corrected the spelling errors you point out.

·         “In Permutation Entropy (PE), sometimes an oversampling parameter…”

o   We have now included this remark explicitly in Section 2.1.1.

·         “After line 79, it would be useful to comment on the potential implications of limiting the series to k=1.”

o   The constrain for the appropriate choice of time scale “m” for a proper MPE analysis are now included in Section 2.1.2. Since for real applications, “m” must be small compared to the signal length “N”, the choice for k=1 is justified. We also added some remarks regarding this choice directly in line 79 (now it is line 98 after editing).

Best regards,

Antonio Dávalos

Reviewer 2 Report

In this interesting paper, authors explore the statistics of the MPE, and especially the estimator’s variance. I found the paper interesting, and especially well written. I have just one major comment (although not very major), and a few suggestions.

And the end of the paper, I got the feeling of “and therefore?”. That is, it would be nice if, in the conclusions, authors could give some advice to the practitioner using MPE. Should one be careful with short time series, but only when the resulting entropy is very low / high? Do results imply that shorter time series (shorter than d!) can be used, provided the probabilities of patterns are similar? Or maybe the opposite: if patterns are not equally probable, one should go beyond d!. I think that some conclusions of this kind could improve the “applicability” of the paper.

Line 41. “this results” -> “these results”

Line 78. “that t frac Nm” I guess this should be N/m

Page 6. I think Fig 1 is not references anywhere in the text…

Line 131. “the simulations graph have the theoretical graph have the same shape” -> “the simulation and theoretical graphs have the same shape”

Page 8. The lines of Fig 2a are difficult to see, especially the yellow one (and especially when printed). It might be useful to increase the contrast, or make the lines thicker.

One last comment, albeit none of my business. The authors have acknowledged a CONACyT project (thus Mexican), but all authors are from France… Is this OK? (Again, none of my business, but it just stroke me as odd, and maybe a mistake)

Author Response

Thank you very much for your comments and suggestions. We increased the thickness of the lines in the figures and corrected the grammar mistakes.

·         And the end of the paper, I got the feeling of “and therefore?”.

o   We have developed further in the comments regarding the implications of the results obtained. Figure 2a (Now Figure 3a) implies that the variance is heavily dependent on the pattern distribution, and thus, in the MPE measurement itself. For entropy values close to the maximum and minimum, the variance is very small. On the other hand, when the normalized entropy is close to 0.4, the variance is maximum. This effect comes from the pattern distribution, and it is independent of the signal length and the time scale selected. Therefore, this must be taken in consideration when comparing two different MPE values, as they will have different variance.

o   Also, when the MPE is near its maximum, the variance linear approximation does not hold, as the term l^T * Sigma_p * l is almost zero. For very high MPE values, the quadratic approximation is necessary. For any other MPE value, the linear approximation is accurate, even for entropy values close to zero (heavily uneven distributions)

We have included this remarks both in the Discussion and the Conclusion sections. We hope this can better comment on the implications of the use of the MPE statistic.

·         The authors have acknowledged a CONACyT project (thus Mexican), but all authors are from France… Is this OK?

o   This is indeed correct. I (Antonio Dávalos) am a Mexican researcher working in the University of Orléans.

Saludos cordiales,

Antonio Dávalos

Reviewer 3 Report

Summary:

The authors propose to extend the results on the analysis of the multiscale permutation entropy (MPE) by developing an explicit expression for the MPE variance as a function of time scale and ordinal pattern distribution. They also claim a formulation of the MPE Cramér-Rao Lower Bound (CRLB) to test the efficiency of their results.

Comments

There are two major issues that require special attention: first with respect to validity of the use of MPE for complexity analysis and the other with regard to the formulation steps by the authors. These points are elaborated below.

Line 180-181: “This leads to a less precise MPE estimation, implied by the increased variance measurement.” – This is not just a matter of “less precision” but in fact the main concern about the validity of application of MPE for complexity analysis of time series. To clarify my point, let’s consider an extreme case in which the actual time series is divided into two windows. Now, based on the central limit theorem the average of the averages of these two windows perfectly estimates the expected value of the original time series. However, their respective values might either be very close to each other (and therefore very close to the actual time series mean) or very different. In the case of the former we will see the entropy vanishes as the binning of these values results into them being grouped together. If they are very different, they will be assigned to two equally weighted bins which will result in maximum entropy (i.e., a binary entropy in this case). The first case indicates a decrease and the latter sets an upper bound (i.e., maximum entropy).  On the other hand, considering the results in this manuscript that suggest the increase in variance of the estimator by “m” points at a concerning issue with MPE which can be understood intuitively: the larger the window size the more probable becomes for the symbols to be repeated more frequently and hence their calculated entropy will increase. I strongly believe that these results must be contrasted against such measures as multiscale entropy (MSE) while using canonical data (e.g., 1/f frequency, Gaussian white noise, etc.) whose dynamical behaviours are well understood to first and foremost justify the validity of MPE for analysis of the signal complexity.

Lines 61-62: “The major disadvantage in PE involves the length of the signal, where we need N >> d! .” - Although this inequality is well stated, general values for “d” are 3 and 6 (multiples of three). For 6! = 720 any device with only a 10Hz sampling rate would need 72 sec. to collect this data. Unless we opt for larger values, say, 9 (9!=362880) , length is of a less concern. I think the issue with PE that requires further scrutiny is how it can be interpreted as a measure of signal dynamics since its calculation ignores the temporal information of the symbolic variation in the time series. A remedy to this which has shown to be valuable (at least within the field of neuroscience) has been to use the symbolic transformation of the signal directly (refs. (a) and (b)) (such a discretization has shown to increase the signal’s information content ref(c)) and then maximize the entropy (ref (d)).
(a) T.O. Sharpee, Optimizing neural information capacity through discretization. Neuron, 94, 954-960 (2017).
(b) J.R. King, J.D. Sitt, F. Faugeras, B. Rohaut, I. El Karoui, L. Cohen, L. Naccache, S. Dehaene, Information sharing in the brain indexes consciousness in noncommunicative patients. Current Biology, 23, 1914-1919 (2013)

(c) S. Moguilner, A.M. García, E. Mikulan, E. Hesse, I. García-Cordero, M. Melloni, S. Cervetto, C. Serrano, E. Herrera, P. Reyes, D. Matallana, Weighted Symbolic Dependence Metric (wSDM) for fMRI resting-state connectivity: A multicentric validation for frontotemporal dementia. Scientific Reports, 8, 11181 (2018).

(d) S. Laughlin, A simple coding procedure enhances a neuron’s information capacity. Zeitschrift für Naturforschung, 36, 910-912 (1981).

MPE performs the coarse-graining as it is done in MSE. This begs the question that how these two differ. In particular, MSE has shown to be less sensitive to signal length (refs. (a) and (b)) and its behaviour can be interpreted in terms of power law and the signal frequencies (refs. (c) and (d)). How would these interpretations lend themselves to symbolically transformed and segmented data in which the temporal information are quite completely lost (due to double coarse-graining; once for computing the symbols and the other through the “M” part of MPE)? The difference between the coarse-graining by MSE and the one in MPE is that the former brings the repeated patterns and the trends in data closer to each other every time it averages the time series within the given window size (which is to say that it reduces the temporal resolution in a controlled manner) which makes the detection of their presence more tractable and as a result it either reduces (in the presence of such patterns) as we move toward higher scale or tends toward a straight line (what has been attributed to the presence of power law) that implies (relatively) no loss of complexity between the consecutive coarse-grained data.

Sokunbi, M.O., 2014. Sample entropy reveals high discriminative power between young and elderly adults in short fMRI data sets. Frontiers in neuroinformatics, 8, p.69.

Courtiol, J., Perdikis, D., Petkoski, S., Müller, V., Huys, R., Sleimen-Malkoun, R. and Jirsa, V.K., 2016. The multiscale entropy: Guidelines for use and interpretation in brain signal analysis. Journal of neuroscience methods, 273, pp.175-190.

McDonough, I.M. & Nashiro, K., Network complexity as a measure of information processing across resting-state networks: evidence from the Human Connectome Project, Frontiers in Human Neuroscience, 2014, 430 8, 409.

Takahashi, T., Cho, R.Y., Murata, T., Mizuno, T., Kikuchi, M., Mizukami, K., Kosaka, H., Takahashi, K. & Wada, Y., Age-related variation in EEG complexity to photic stimulation: aa multiscale entropy analysis, Clinical Neurophysiology, 2009, 120, 476–483.
Line 71: “The calculated MPE from the signal depends directly on the calculated pattern probabilities …” This does not only apply to PE/MPE but any estimator that compute the entropy through one or another types of discretization of the signal. For instance, using non-parametric approaches to compute the entropy of continuous random variable may rely on such algorithms as k-nearest-neighbor (KNN) which is to avoid the bias imposed by using the binning-based approaches. Nevertheless, they both need computing the probabilities that are computed through discretization.

Equations (A13) through (A15): Considering the general form of the entropy for discrete random variable, we have H(p(x)) = -\sum_x p(x) log(p(x)) which has a derivative: \partial H(p(x))/ \partial p(x) = -1 – log(p(x)) and not the one shown in (A13). In particular, in the case of (A13), it appears to me that the derivative is suffering from a misinterpretation of factoring the p_d!. In general, if Y = x +/- y then \partial Y/ \partial x = 1 and nothing from y part since it has no “x” term in it (in other word, the p_d! ln p_d! part would be canceled). Therefore, I don’t quite follow the first two expressions in (A13) that lead to the final derivative as introduced by the authors. Even if we were to assume that the step taken by the authors is legitimate, then there is a problem with the way they deal with the logarithmic term. Derivative of a logarithmic function F(x) = log(g(x)) is: \partial F/ \patrial x = (\partial g/ \partial x) / (ln(b) g(x)) where “b” is the base of the logarithm.

Lines 147-148: “This means that we are only interested in the region where m/N is close to zero, where the high order terms of (18) vanish.” This is not a conclusion/observation based on the authors’ analysis but in fact a necessary condition/assumption in order to justify getting from equation (18) to equation (19). Unless the assumption of “m/N” is introduced in advance, the authors cannot simply justify cancelling the last two terms in equation (18) and move straight to equation (19). In fact, this can be seen in line 86 where “0 < m/N < 1” is mentioned. Now, consider m/N = .99 and the contribution of the last two terms is substantial.

With regard to equation (17): Following equation (6), we have: Y_i = N/m \hat{p_i} = N/m p_i + \delta Y_i => \delta Y_i = N/m\hat{p_i} – N/m p_i => \delta Y_i = N/m(\hat{p_i} – p_i). If so, how the term [\delta \bold{Y}^circ2] is replaced by (p – p^circ2)  (i.e., the power of 2 only on the second term in the parenthesis) when “\delta Y_i = N/m(\hat{p_i} – p_i)” and the power of 2 acts on all components of \delta\bold{Y}^\circ.

Line 84: “Since (18) is a cubic polynomial the variance function is guaranteed to have a zero at m/N = 0.”: First, I don’t see a cubic polynomial but (m/N)^3. Second, every term in this equation has a m/N coefficient which means they would all be cancelled once multiplied by m/N = 0. And then the claim that “the variance function is guaranteed to have a zero at m/N = 0” can only happen when m = 0 (if N = 0 then the expression is undefined, in addition to the fact that there would be no time series to begin with). Depending on how to interpret m = 0, there could be two possibilities: m = 0 implies using the entire data  and this in turn means \hat{H} = H since we are calculating it based on the full time series. And if m = 0 implies no operation (that is to say that m =N means using the full time series) then there is nothing to be computed and hence no variance at all. In addition, the claim is self-explanatory since m/N = 0 will set the whole expression in (18) to zero. Second, what is the significance of such an observation. I assume this is a statement for the authors to justify that why “we” are interested in m/N -> 0. However, such a condition is a necessary assumption (as mentioned above) for the authors to move from (18) to (19) and does not verify the cancellation of the last two terms in (18).

Equation (8): There is also a concern with regard to the way that the authors (based on their previous study) formulated the expected value of the estimator. In this equation, m is the window size. Now, as m tends to N then estimated value of H must in principle to H since m goes to N implies that we are getting closer and closer to the full time series. However, equation (8) does not really reflect this since the second term on the right will not go away. In addition, consider the case where m = N (i.e., using the whole time series). In such a scenario (d!-1)/2N * m = (d!-1)/2 (when m = N \hat{H} = H i.e., the best estimate of H for a data is using the full data and computing the H for it) in which case unless H > (d!-1)/2 the E[\hat}] < 0 (for instance, if d = 3, then H must be >= 2.5 which is not necessarily warranted). However, the entropy of a discrete random variable cannot be negative (always >= 0 although it might be negative in the case of differential entropy).

Equation (16): p appears to be a vector in which case “pp^T” gives its norm (i.e., a single quantity). How does this lead to covariance matrix? Moreover, diag(p) is a diagonal matrix (i.e., all elements except diagonals = 0). Regardless of what the resulting matrix after subtracting the norm from the diagonal matrix represent,  (diag(p) – pp^T) yields to “-m/N\Sigma_p” (the result in this equation appears to miss a negative sign).

Line 136-139: p_i = 1/d! corresponds to the uniform distribution that results in maximum entropy (also mentioned by the authors). I don’t quite understand then why seeing that \hat{H} converges to H i.e., its variance tending to zero as we approach the maximum entropy should be considered “interesting” or surprising. This is what is expected by the maximum entropy theorem that roughly mentions when we don’t know the underlying distribution then we should pick the one that maximizes the entropy (in the case of discrete random variable that distribution is uniform distribution).

Figure 2: Isn’t this expected, given the setting? It is a case of binary entropy and therefore one side of the plot corresponds to the case of p > (1 – p) and the other side to the opposite case. Therefore, the plot will be symmetric on the two side of the point p = (1 - p) = .50 that is where the entropy is maximum (i.e., variance goes to zero in case of this plot).

Line 202: “We note that Y is composed of a constant part np_i, and a random part \delta Y_i” – I don’t understand how “np_i” is considered a constant by the authors. Yes, it is the expected value of binomial distribution (we replace p_i with p) but that doesn’t mean it is a constant. “n” is but “p_i” varies by the frequency of the symbol “i.” “p_i” is estimated by the number of symbol “i” with respect to the sum of frequencies of all symbols.  

Line 78: What is “tfracNm”? I suspect it is due to missing a “\” in the LateX script and it should read “t \times N/m” in which case I don’t see “t” being referenced/defined before this line.

Based on (A14), \bold{1} is nothing but a vector with all its elements = 1. Then what is the meaning of multiplying by \bold{1}^T (e.g., (18))?

Some minor comments include:

In Abstract (and more general throughout the text), please avoid switching between tenses. A few examples in the Abstract are:“… we expand” and then line 5: “We also formulated …” which is then switched again in line 7: “… variance is also tested …”

There is no need for capitalizing “entropy,” “permutation entropy,” and “multiscale entropy.”
Line 17: “… has been used as the measure …” should read “ .. as a measure …”
When referring to equations in the text, it is helpful if you use the term “equation” to prevent readers confusing them with the cited references.
Line 41: “This results” should read “These results”
Line 46: “The document … ” should read “The remainder of this article”
Line 65: “MPE consists on ..” should read “… consists of …”
Equation (11): \hat{p} is not explicit in (10) and does not need to be defined in (11).
Line 95: “for generate” should read “for generating” or “to generate”
Line 101: “without loss of generality” – Does this mean on the variance of Gaussian noise?  If so, I don’t think that increasing the noise variance does not have any effect on the trend of the time series. Please clarify.

Author Response

Thank you very much for your comments and suggestions. We have carefully reviewed our work to further illustrate and develop our findings in this manuscript. We will address your concerns regarding our work as follows:

·         Line 180-181: “This is not just a matter of “less precision” but in fact the main concern about the validity of application of MPE for complexity analysis of time series.”

o   It is true that, if we divide the signal in two windows, it will lead to a highly unstable entropy measurement. Nonetheless, the construction of a histogram with two data points will not be meaningful to estimate the probability distribution. Unless the original signal is very small, this problem will only arise when we choose a window size that is too large for the analysis. In the original formulation of Permutation Entropy, Bandt and Pompe (a) do address this limitation.

o   The validity of the MPE is not addressed in this work, since it has been extensively tested by other authors (b) both with canonical data and real biomedical signals.

o   It is indeed intuitive that a larger scale will yield to an increase in the MPE variance. Nonetheless, this increase has not been properly characterized before.

§  (a) Bandt, C.; Pompe, B. Permutation Entropy: A Natural Complexity Measure for Time Series. Physical Review Letters 2002, 88, 174102. doi:10.1103/PhysRevLett.88.174102.

§  (b) Azami  and  J.  Escudero.  Improved  multiscale  permutation entropy for   biomedical   signal   analysis:   Interpretation   and   application  to electroencephalogram  recordings.  Biomedical  Signal  Processing  and Control Vol. 23 P28-41. 2016

·         Lines 61-62: “Unless we opt for larger values, say, 9 (9!=362880) , length is of a less concern.”

o   In general, the coarse-graining procedure will produce signals that have a reduced length of N/m. This means that the length of the signal eventually becomes a concern, even for a small embedded dimension d.

·         “MPE performs the coarse-graining as it is done in MSE. This begs the question that how these two differ.”

o   While both MPE and MSE apply their respective calculations to the same coarse-grained signals, MPE is calculated based on the ordinal pattern of segments, while the MSE uses the Sample Entropy (SampEn), in which the difference between data values within a segment is relevant. Although MSE is less sensitive to signal length, it is highly sensitive to the signal amplitude, reporting an increase in entropy as time scale increases, and the data points come closer together. MPE is invariant to this decrease in amplitude, as it registers the order of the elements inside the window of size d even before the coarse-graining procedure. It also detects patterns that are similar in shape, but separated in the ordinate axis.

o   Both MSE and MPE do this “double coarse graining”. Both SampEn and PE require the definition of a segment length for comparison (even though they are performing different calculations). Both MSE and MPE perform the same coarse-graining procedure described in Section 2.1.2.

·         Line 71: “The calculated MPE from the signal depends directly on the calculated pattern probabilities …” This does not only apply to PE/MPE but any estimator that compute the entropy through one or another types of discretization of the signal.

o   The original intent of this remark was to contrast with the case of the MPE bias, which is independent of the pattern distribution. I agree this remark is confusing, so we corrected it accordingly.

·         Equations (A13) through (A15)

o   Since all pattern probabilities must add to 1, we must apply the restriction in Equation A11, where p_d! depends of all the other probabilities. This term is not cancelled by applying the derivative respect to any p_i <>p_d!

o   The term p_d! ln p_d! –p_j ln p_j is indeed wrong. It should read ln p_d! –ln p_j. Fortunately, this is a typing error, and does not disrupt the rest of the calculations.

o   From A11, the derivative of p_d! with respect to any other p_j is -1, which only changes the sign of the logarithm. This is why ln(p_d!) is positive in A13

·         Lines 147-148: “This is not a conclusion/observation based on the authors’ analysis but in fact a necessary condition/assumption in order to justify getting from equation (18) to equation (19).”

o   I agree the parameter constrains for m and N are not explicitly laid down. We have edited the manuscript accordingly, explaining this necessary condition in Section 2.1.2. , and the proper justification during the MPE variance derivation in Section 2.2

o   Nonetheless, the term m/N indeed needs to be small, for practical reasons. A coarse-grained segment with 99% the size of the signal length (m/N=0.99) will not lead to a meaningful analysis.

·         With regard to equation (17):

o   By taking each element of the vector \Delta \bold{Y}

§  \Delta Y_i = (N/m)*( \hat{p_i} – p_i )

§  \Delta Y_i^2 = (N/m)^2*( \hat{p_i} – p_i )^2

§   

§  \Delta Y_i = (N/m) ^2*( \hat{p_i}^2  – 2 *p_i*\hat{p_i} + p_i^2)

§  E[\Delta Y_i] = (N/m) ^2*( E[\hat{p_i}^2 ] – 2 *p_i*E[\hat{p_i}] + p_i^2)

§  E[\Delta Y_i] = (N/m) ^2*( E[\hat{p_i}^2 ] – p_i^2)

§  Now we calculate E[\hat{p_i}^2 ] as follows:

§  E[\hat{p_i}^2 ] = (m/N) ^2*E[y_i^2 ]

§  E[\hat{p_i}^2 ] = (m/N) ^2*( var(y_i) + E[y_i^2 ] )

§  E[\hat{p_i}^2 ] = (m/N) *p_i - (m/N) *p_i^2 + p_i^2

§  Therefore:

§  E[\Delta Y_i] = (N/m) ^2*( (m/N) *p_i - (m/N) *p_i^2 + p_i^2 – p_i^2)

§  E[\Delta Y_i] = (N/m) *( p_i - p_i^2)

§  Since this is true for each value of  \Delta Y_i, we can group these results in the following vector

§  E[\Delta \bold{Y_i}] = (N/m) *( \bold{p} - \bold{p}^circ2)

§  Equation 17 follows from this result. I hope this clarifies this calculation.

·         Line 84: “First, I don’t see a cubic polynomial but (m/N)^3…”

o   We have clarified in the manuscript that this equation is a cubic polynomial respect to (m/N)^3

o   Indeed, the part where we mention (m/N)=0 is not relevant to our analysis. We have promptly erased this part from the manuscript.

·         Equation (8)

o   This result from a previous study also requires that m/N to be small. As mentioned before, we have now made this clarification in Section 2.1.2.

o   The comments we made in “Lines 147-148” also apply here.

·         Equation (16): “p appears to be a vector in which case “pp^T” gives its norm”

o   The term “pp^T” is not the norm, but a matrix which contains all the cross-term multiplications of the elements of p. Since p is defined as a vertical vector, “pp^T” yield to this matrix, while “p^T*p” is the dot product of p

o   Based on this, we have clarified that all vectors in this manuscript are vertical/column vectors.

·         Line 136-139: p_i = 1/d! corresponds to the uniform distribution that results in maximum entropy (also mentioned by the authors):

o   We have further expanded our comments regarding the properties of the MPE variance. It is to be expected that the uniform distribution yields to the maximum possible entropy. For signal measurements, however, we do not have access to the true pattern probabilities p_i. Only the probability estimators \hat{p}_i are available. Therefore, we will still have a variance, albeit small, even when the true parameters from the signal are uniform.

·         Figure 2: “Isn’t this expected, given the setting?”

o   The symmetry is indeed expected, but not this particular structure. We have expanded our comments in this regard.

·         Line 202: “I don’t understand how “np_i” is considered a constant by the authors”

o   We now changed the word “constant” with “deterministic”, to better convey the idea that n*p_i is not random.

·         Line 78: What is “tfracNm”?

o   This is a typing error. We have now corrected it.

·         “Based on (A14), \bold{1} is nothing but a vector with all its elements = 1. Then what is the meaning of multiplying by \bold{1}^T (e.g., (18))?”

o   \bold{1} is indeed a vector of ones. When we use the term \bold{1}^T*\bold{p}, for example, we are calculating the dot product of vector \bold{p} and a vector of ones. This is a convenient way to write the sum of all elements of \bold{p} in vectorial form, as

§  \bold{1}^T*\bold{p} = p_1 + p_2 + … + p_d! = \sum(p_i)

We took your comments regarding grammar and structure in consideration, and edited the manuscript accordingly. We expect to have addressed all your concerns regarding our work.

Best regards,

Antonio Dávalos

Round 2

Reviewer 1 Report

I appreciate the authors' effort to address the comments.

Reviewer 2 Report

Authors have addressed all my comments, so I recommend the publication of this paper.

Reviewer 3 Report

Author: “This result from a previous study also requires that m/N to be small. 

Reviewer: As mentioned before, we have now made this clarification in Section 2.1.2.” Your mention of small “m” (which is not clear in the Abstract, since your claim there is very general), yet a proof must withhold for trivial cases. if you set m = N which is a legitimate sanity check for this equation, it means we are considering the entire signal which is the original premise of actual PE then E[H] = H is a MUST. How do the authors justify that their expected H is less than actual entropy when the same signal is used?!

Authors:  We have clarified in the manuscript that this equation is a cubic polynomial respect to (m/N)^3

Reviewer: Equation (19) is not “cubic polynomial.” Degree of a polynomial is with respect to its variable and not the power of its coefficients.

Equation (A13): It is not just the ordering of the terms in this equation but the the actual derivative is not correct. As I mentioned in my first comments, you are taking derivative of a function that has a logarithmic term. Logarithm itself is a function and ha its own derivative rule. Also, entropy has its own closed form derivative. Please refer to my previous comment on this.

Authors: “We now changed the word “constant” with “deterministic”, to better convey the idea that n*p_i is not random.” 

Reviewer: A statement that has a probability as one of its parameter is neither “constant” nor “deterministic.” or otherwise what is point of talking about the probability in the first place? This statement particularly (as I mentioned in my previous comments) closely relates to the expected value of geometric distribution. For instance, if we toss a coin N time how long would it take to see the first “1”? I don’t think anyone can claim this problem has a “constant” or “deterministic” answer.

In Page 5: “Since we are still working in the region where m/N is close to (but not including) zero,” : How do the authors define such a limit? What is the lower bound that they are considering? It is a mathematical proof that they are proposing and as such it needs its requirements and statements well-formed. For instance, is .1 small enough? Then 1/10, 10/100, 100/1000, 1000/10000 all yield the same value. Which one of these is considered to be a good setup for m and N? 

Authors: “Nonetheless, the term m/N indeed needs to be small, for practical reasons. A coarse-grained segment with 99% the size of the signal length (m/N=0.99) will not lead to a meaningful analysis.” 

Reviewer: a proof must withhold its boundary cases. Assume N = 1000, now assume I trim it to N = 990. Should this trimming make a significant difference? m/N = .99 corresponds to such a hypothetical scenario? I don’t see how PE of signal of length 1000 will be drastically different from its PE when its length is shortened by 10 data points. 

Regarding your response

 “In general, the coarse-graining procedure will produce signals that have a reduced length of N/m. This means that the length of the signal eventually becomes a concern, even for a small embedded dimension d.” : This does not have anything to do with the parameters that one uses for PE/MPE but whether the actual signal is relevant for such analyses.

While both MPE and MSE apply their respective calculations to the same coarse-grained signals, MPE is calculated based on the ordinal pattern of segments, while the MSE uses the Sample Entropy (SampEn), in which the difference between data values within a segment is relevant.” : MSE also considers patterns although not in terms of templates as it is the case for PE. There consecutive signals are compared with respect to some threshold (e.g., some ratio of the signal variance) to see if they continue to stay within the given threshold. 

reporting an increase in entropy as time scale increases, and the data points come closer together.” Regardless of the type of measure, this statement is invalid. “the data points come closer together” implies they become more similar which means entropy must decrease. 
